# Revealing hidden interlayer excitons in 2D bilayers via hybrid molecular gating

Sviatoslav Kovalchuk ®[1]✉, Kyrylo Greben ®[1], Abhijeet M. Kumar[1], Simon Pessel[1], Jan Soyka[2], Qing Cao[2], Kenji Watanabe ®[3], Takashi Taniguchi ®[3], Dominik Christiansen[4], Malte Selig[4], Andreas Knorr ®[4], Siegfried Eigler ®[2] & Kirill I. Bolotin ®[1]✉

Heterostructures of molecules and two-dimensional materials feature emergent properties not seen in their individual components. Here, we study excitons in bilayer transition metal dichalcogenides exposed to an intense electric field produced by charge transfer from proximal molecules. Our approach allows for reaching an electric field strength of 0.35 V nm$^{-1}$, up to a factor of two higher than previously achieved in purely solid-state gated devices. Under this field, inter- and intralayer excitons are brought into an energetic resonance, allowing us to explore a new physical regime. We detect a previously unseen interlayer exciton that only becomes visible at high electric field through hybridization with the intralayer A exciton. Moreover, the system experiences an ultra-strong Stark splitting of > 350 meV with exciton energies tunable over a large range of the optical spectrum, holding potential for optoelectronics. Our work paves the way for using strong electric fields to study new physical phenomena and control exciton hybridization in 2D semiconductors.

Interlayer excitons (IX) in bilayer transition metal dichalcogenides (2L-TMDs) are Coulomb-bound pairs of electrons and holes with an out-of-plane dipole moment. Compared to intralayer excitons, the electron-hole separation in IXs is larger and the oscillator strength lower[1–4]. As a result, IXs feature lifetimes in the tens of nanoseconds[4–6] and diffusion lengths up to microns[7,8], much higher than their intralayer counterparts. These properties led to an explosion of interest in IXs in fields such as excitonic transport[8,9], Bose-Einstein condensation[10–12], excitonic insulators[13,14], and quantum simulation[15]. In homobilayers, the distinguishing property of IXs is their coupling to intralayer excitons resulting from interlayer hole tunneling[3,16–19]. This mechanism leads to a tunable enhancement of the oscillator strength of IXs in some 2L-TMDs, e.g., 2L-MoS$_2$, allowing their observation via optical absorption spectroscopy[16,17,20]. Finally, IXs exhibit a Stark splitting in electric fields oriented perpendicularly to the plane of the material due to their out of plane static dipole moment[4,16,17,20–22]. As a result, the energy position,

oscillator strength and coupling strength to other excitonic species can be tuned by an electric field.

A perpendicular electric field in 2L-TMDs is conventionally applied in a dual-gated field effect transistor geometry. Electrostatic gates consisting of a dielectric (e.g., hBN or SiO$_2$) and a conductor (e.g., gold, Si or graphene) are assembled on both sides of the 2L-TMD. In such a configuration, the difference between gate voltages applied to the top and bottom conductors controls the field across the material, while the sum of gate voltages controls the carrier density and Fermi energy[23]. Generally, the strength of the perpendicular electric field controls the energy splitting between IXs with oppositely oriented dipole moments (denoted IX$_+$ and IX$_-$). The maximum reported splitting in conventional dual-gated devices[17,24–26] is in practice limited by the breakdown of the dielectric material. At the point of dielectric breakdown of hBN, the electric field inside a 2L-TMD reaches ≈ 0.2 V nm$^{-1}$. Assuming an interlayer exciton dipole moment of 0.6 e · nm, this

[1]Physics Department, Freie Universität Berlin, Berlin, Germany. [2]Institute of Chemistry and Biochemistry, Freie Universität Berlin, Berlin, Germany. [3]National Institute for Materials Science, Tsukuba, Japan. [4]Physics Department, Technische Universität Berlin, Berlin, Germany. ✉e-mail: kovalchook@gmail.com; bolotin@zedat.fu-berlin.de

electric field corresponds to a Stark shift of 120 meV, smaller than the separation between A and B excitons for most 2L-TMDs, e.g., 240 meV in WSe$_2$. As a result, it is challenging to explore the fascinating regimes of hybridization of IXs with both of these intralayer excitons.

While an order of magnitude higher electric field has been recently generated using ionic liquids[23,27], that approach is so far limited to room temperature and incompatible with optical measurements.

Here, to study the regime of tunable coupling between IXs and other excitonic species, we overcome the limits of solid-state gating technologies. We develop a hybrid molecular gating approach that allows the generation of an electric field of $> 0.35$ V nm$^{-1}$ (displacement field $> 2.2$ V nm$^{-1}$), nearly doubling the previous limit. A Stark splitting of $> 350$ meV allows us to discover a new high energy interlayer exciton in bilayer MoS$_2$, labeled IX$_2$, by hybridyzing with X$_A$. In bilayer MoSe$_2$, meanwhile, the high electric field reveals a new dark interlayer exciton state.

## Device concept/Evaporation Technique

To overcome the limits of conventional gating, we add layers of charges next to the 2L-TMD (Fig. 1a). The top layer consists of acceptor molecules with charge density $\sigma_t$. We use either 2,3,5,6-Tetrafluoro-7,7,8,8-tetracyanoquinodimethane (F$_4$TCNQ), a well-known commercially available molecular acceptor with electron affinity of 5.3 eV[28–34] or hexacyano-trimethylene-cyclopropane (CN6-CP), a tailor-made molecular acceptor with high electron affinity of 5.94 eV[35–37] (details

of synthesis and characterization are given in SI). The bottom layer of charges originates from the donor states already present at the interfaces between the TMD and the SiO$_2$/Si stack. These states, with charge density $\sigma_b$, arise due to a combination of photodoping[38–40] and interface charge trapping[41,42].

A large electric field inside the 2L-TMD is generated due to a high density of charges in the top layer compensated by the combination of bottom layer and electrostatic gate (voltage $V_G$). The subtle but critical aspect of this molecular gating approach is that the charges on both sides of the device are localized rather than free. These charges cannot result in persistent currents, which is one of the main mechanisms that leads to dielectric breakdown. Using a three-capacitor electrostatic model (Fig. 1a), we determine the electric field in the TMD layer as (SI note S1):

$$F_Z \approx \frac{1}{2\varepsilon_0\varepsilon_{TMD}}(\sigma_t - \sigma_b - V_G C_G),\tag{1}$$

where $C_G$ is the areal capacitance between the bilayer and the Si gate. The formula shows that the electric field inside the 2L-TMD can exceed the field inside the SiO$_2$, the last term inside the parenthesis. We note that the gate voltage $V_G$ controls the chemical potential alignment between the 2L-TMD and the molecules and hence determines the charge density $\sigma_t$ in them (Fig. 1a). When $V_G$ is gradually increased, both the electric field and the Fermi energy increase with it, up to the point when the Fermi energy reaches the minimum of the conduction band

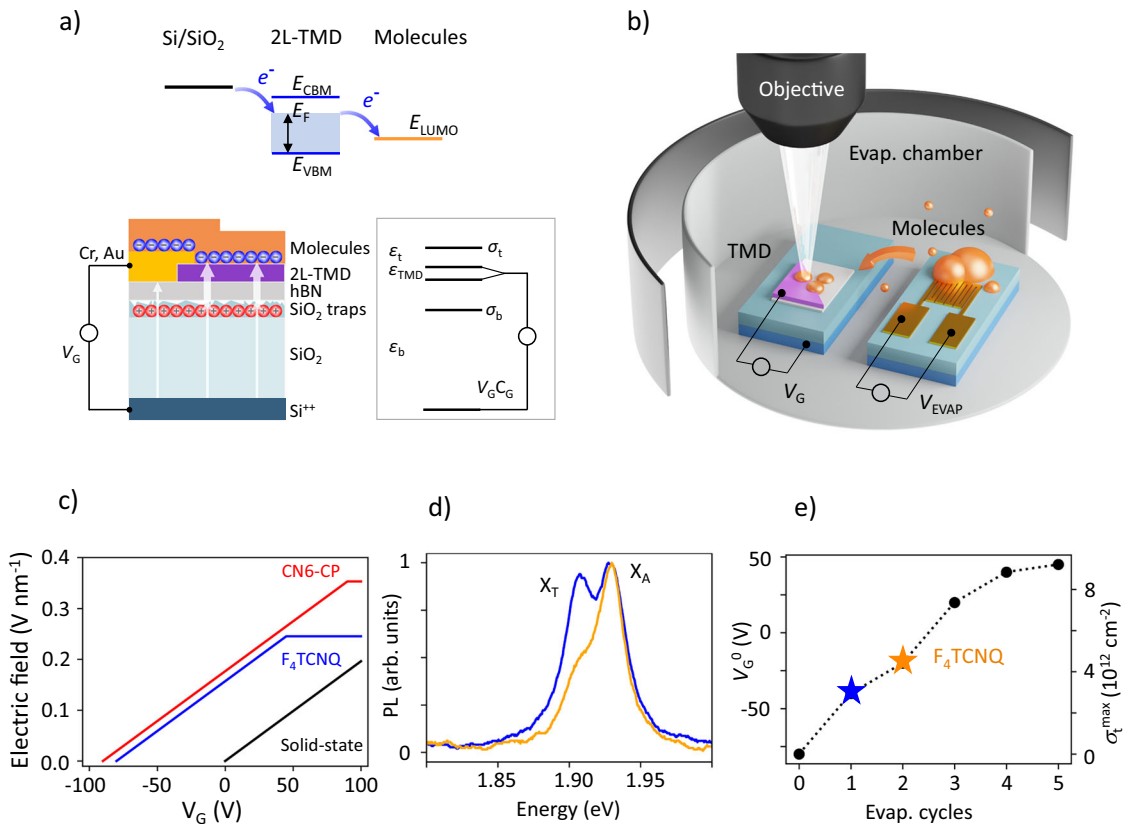

**Fig. 1 | Hybrid molecular gating. a** Device schematic of a 2L-TMD with two layers of charge, and an equivalent 3-capacitor circuit. The top inset illustrates the energetic alignment between SiO$_2$/Si gate, 2L-MoS$_2$ and F$_4$TCNQ. **b** Evaporation chamber setup schematic. Organic molecules are deposited onto a TMD in situ by controllably heating the coil on a separate chip. **c** Calculated electric field dependence on the gate voltage for the device described in the text based on the molecular dopant (blue and red) versus solid-state only device (black).

**d** Photoluminescence spectra of 2L-TMD before (blue) and after (orange) evaporation of F$_4$TCNQ, at fixed $V_G = $ -20 V. The decrease of the trion peak X$_T$ intensity indicates that 2L-MoS$_2$ becomes more neutral due to charge transfer. **e** The smallest voltage at which the trion feature is visible in the optical spectra ($V_G^0$) vs. the number of the evaporation cycle. The maximum charge transfer density in the molecules ($\sigma_t^{max}$) is shown on the right axis. Molecular density at points shown as blue and orange symbols correspond to the curves of the same color in (**d**).

of one of the layers. In this situation, the screening due to free carriers induced in the 2L-TMD limits additional field increase.

To illustrate the utility of the molecular gating approach, we modeled the electric field vs. $V_G$ inside 2L-TMD in Fig. 1c (see SI note 1 for modeling details). We consider three cases: devices based on either F$_4$TCNQ or CN6-CP molecules as well as a double-gated device based on top and bottom dielectric-based gates. The maximum electric field in the latter device is limited to 0.2 V/nm by the dielectric breakdown of the dielectric. In contrast, for the case of the CN6-CP device, the field strength exceeds that value almost by a factor of 2.

We control the maximum carrier density in the molecular layer $\sigma_t^{max}$ - and with it the maximum $F_Z$ - during the experiment using a newly developed in situ molecular evaporation technique. The approach works by applying short pulses of current to a micro-fabricated coil on a separate chip loaded with molecules. That chip is placed close to the measured sample inside the cryostat (Fig. 1b, details in *Methods)*. The temperatures of both chips are monitored by microfabricated thermometers. Even when the temperature of the evaporator chip reaches 400 K, the sample remains at near Helium temperature (Si Fig. 10). This approach adds more flexibility compared to traditional deposition techniques[33,43–45], enables the evaporation of oxidizing molecules, and allows precise control of the surface

coverage, which could be even more crucial when using other organic molecules, for example dyes[46].

We confirm molecular deposition by recording photoluminescence (PL) and reflectivity spectra during cycles of molecular evaporation (Fig. 1d, e). In PL measurement, we observe spectral changes in the region of the intralayer neutral exciton (X$_A$ at 1.94 eV) and intralayer trion (X$_T$ at 1.91 eV). The gradual decrease of X$_T$ brightness is consistent with the change of the carrier density in the sample due to the deposition of acceptor molecules onto the 2L-MoS$_2$[33]. The quantitative analysis of an exemplary F$_4$TCNQ sample indicates that $\sigma_t$ can be gradually increased from 0 to $9 \times 10^{12}$ cm$^{-2}$ in that device via molecular deposition (Fig. 1e).

## Results

### Stark splitting in bilayer TMD systems in linear approximation

We now study the effect of the electric field on interlayer excitons in a CN6-CP/2L-MoS$_2$ device (the data for a F$_4$TCNQ/MoS$_2$ sample are in the SI). Figure 2a shows the map of the second derivative of the reflectivity contrast as a function of $V_G$. We identify the spectral features corresponding to intralayer X$_A$ and X$_B$ excitons (1.93 and 2.10 eV, respectively), at $V_G = -80$ V) as well as interlayer exciton IX$_1$ ( ~ 2 eV at $V_G = -80$ V) that undergoes Stark splitting in a non-zero electric field.

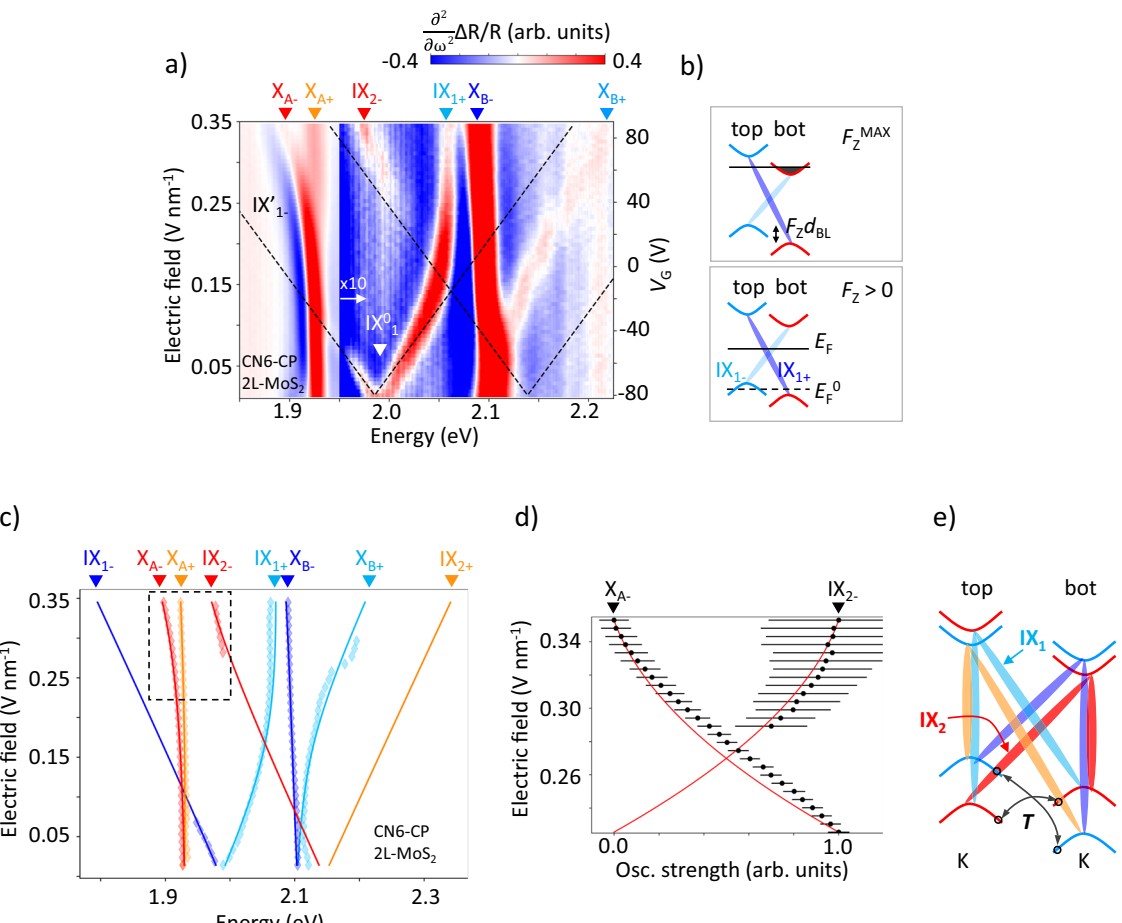

**Fig. 2 | Measuring excitonic response under a strong electric field. a** Map of the second derivative of reflectivity contrast (R$_C = \Delta R/R$) for a CN6-CP/2L-MoS$_2$ sample. The spectra above 1.95 eV are multiplied by 10 to increase contrast. **b** Sketches showing the composition of interlayer excitons for different electric field strengths. IX$_{1+}$ is shown as dark-blue, and IX$_{1-}$ as light-blue. The black line indicates the Fermi level. **c** The dependence of the excitonic peak energies on the electric field extracted from the data in Fig. 2a (diamonds), along with theoretical predictions based on the Bloch equations (lines). Note that the coupling to a new interlayer

exciton IX$_2$ leads to the energy splitting of X$_A$ at high electric field. **d** Normalized oscillator strength of the IX$_{2-}$ and X$_{A-}$ states and corresponding error bars vs. the electric field extracted from the spectra (black dots), and the oscillator strength predicted from the Bloch equations model (red line). **e** The configurations of various interlayer and intralayer excitons in 2L-TMDs at non-zero electric field. The coupling between the excitons sharing electron wavefunctions is mediated by spin-conserving interlayer hole tunneling (arrows). Coupled pairs of inter- and intralayer excitons are marked with the same color.

The interlayer exciton hybridizes with the intralayer excitons at higher field leading to an avoided crossing pattern. We start with a simple model that neglects inter-exciton hybridization, while later confirming the results of that model by taking hybridization into account. For each $V_G$, we determine the Fermi energies of top and bottom TMD layers and find the corresponding electric field strengths (see details on the electrostatic model in SI note S1). We then obtain the expected position of the Stark-split interlayer excitons $E_{IX1\pm} = E^0_{IX1} \pm F_Z d_{BL}$, where $E^0_{IX1} = 1.99$ eV is the known spectral position of $IX_1$ at zero field (denoted $IX^0_1$ in Fig. 2a) and $d_{BL} = 0.6$ e · nm is its dipole moment corresponding to the TMD interlayer distance[1]. The positions of $IX_{1+}$ and $IX_{1-}$ as a function of $V_G$ obtained in that way are shown as dashed lines in Fig. 2a. The left axis shows the electric field obtained from the same model.

At $V_G = -80$ V, where $F_Z \approx 0$, the two interlayer components $IX_{1+}$ and $IX_{1-}$ are close to degenerate. The Fermi energies of the two TMD layers are aligned with the lowest unoccupied molecular orbital ($E_{LUMO}$) of CN6-CP, while $\sigma_t = 0$; the electric fields from $\sigma_b$ and the gate electrode are compensated. In the range of $V_G$ from -80 V to -40 V, the limit of small $F_Z$, spectral shifts of the $IX_{1\pm}$ excitons are linear, matching our simple model. At higher $V_G$, the electric field becomes large enough to bring inter- and intralayer excitons into an energetic resonance[16,17,20,47]. In general, it is known that the interlayer exciton $IX_1$ couples to the intralayer exciton $X_B$ via hole tunneling[1] (arrows in Fig. 2e; $T$ denotes the tunneling strength parameter). This means that the state $IX_1$ partially acquires the intralayer character of $X_B$ and hence deviates from the linear electric field dependence[1,48]. Conversely, the state $X_B$ hybridizes with $IX_1$ acquiring an interlayer character and splits into two components $X_{B+}$ and $X_{B-}$. At $V_G > 0$, a higher-lying Stark-split interlayer exciton ($IX_{1+}$) crosses the position of $X_B$ and the hybridization with it decreases afterwards. The exciton again approaches a linear electric field dependence (dashed line) in the regime of high $V_G$. Finally, above $V_G = 90$ V, the electric field does not increase further as the Fermi energy of the bottom layer reaches the conduction band minimum and free carriers begin to screen the field (Fig. 2b, top).

A striking feature of the high electric field region is the appearance of two new features: an excitonic peak around 1.97 eV and the splitting of $X_A$ into two peaks. We suggest that both features originate from another interlayer exciton $IX_2$ schematically shown in Fig. 2e. This exciton, which can also be labeled as the "interlayer B exciton", acquires its oscillator strength via hybridization with the $X_A$ exciton mediated by hole tunneling, in exact analogy with the well-known brightening of $IX_1$ through hybridization with the $X_B$ exciton (Fig. 2e).

### Coupling between inter- and intralayer excitons
To confirm the nature of the new excitonic state as well as to ascertain the electric field magnitude, we analyze the effect of the coupling between inter- and intralayer excitons. We extract the energies of all excitonic peaks (Fig. 2c, diamonds for CN6-CP/2L-MoS$_2$, data for F$_4$TCNQ/2L-MoS$_2$ are in the SI Fig. 6) vs. electric field and fit them to a model based on the Bloch equations formalism[16] (solid lines, details in SI note S4). This model assumes that $IX_{1-}$ couples to $X_{B-}$ only, $IX_{1+}$ to $X_{B+}$ only, $IX_{2-}$ to $X_{A-}$ only, and $IX_{2+}$ to $X_{A+}$ only and neglects other more complex types of couplings[47,49]. The free parameters of the model are the energies of inter- and intralayer excitons at zero electric field (unperturbed by interexcitonic interactions), dipole moment $d_{BL}$, bottom carrier density $\sigma_b$, hole tunneling strength $T$, assumed same for $X_A$ - $IX_2$ and $X_B$ - $IX_1$ couplings[48] and intralayer excitons polarizability $\beta_Z$ (SI Fig. 8). The modeling result matches the observed positions and amplitudes of all excitonic peaks for both types of samples for all gate voltages (SI Fig. 6, 7). We extract the tunneling strength $T = 37.58 \pm 1$ meV.

From the coupling model, we find that under the maximum electric field the new observed $IX_{2-}$ state is shifted by 152 meV from its position at zero electric field, $E^0_{IX2} = 2.139 \pm 0.002$ eV. This is 168 meV

above $E^0_{IX1}$, a value close to the spin-orbit splitting of the valence band[50]. In the small electric field regime, the state is far from $X_A$, and its oscillator strength, acquired by coupling to $X_A$, is low. In the regime of high electric field strength, hybridization with $X_A$ brightens $IX_{2-}$, allowing its direct observation with spectral position and oscillator strength of the state that matches our model (Fig. 2c, d). The derivative of the reflectivity contrast with respect to $V_G$[51], shown in SI Fig. 13, also reveals features consistent with the positions of $IX_{2+}$ predicted by the model (solid lines in Fig. 2c).

### Dark excitons in 2L-MoSe$_2$
We now turn to another bilayer material from the TMD family, MoSe$_2$. Using the same analysis as above, we characterize excitons in F$_4$TCNQ/2L-MoSe$_2$, Fig. 3a. We find a dipole moment of $d_{BL} = 0.65 \pm 0.02$ e · nm for interlayer excitons and an exciton tunneling strength $T = 44 \pm 2$ meV. The parameter $T$ is 10 meV lower compared to calculations[48,50]. Finally, due to a larger energy difference between $X_A$ and $IX_2$ (371 meV in MoSe$_2$, compared to 236 meV in MoS$_2$) the splitting of $X_A$ is smaller and less pronounced, reaching 6 meV at the highest electric field.

Additionally there is a new feature that was not identified in MoS$_2$ samples. We observe the weak avoided crossing between $X_A$ and $IX_1$, as well as the coupling of $X_A$ to the previously unobserved state 30 meV below $IX_1$ (Fig. 3a, zoomed in region around $X_A$ shown in Fig. 3b). We simulated the absorption spectrum assuming that $X_A$ couples to two interlayer states at the corresponding energies, with coupling strengths of 5 and 10 meV, respectively. This simulation matches the data well (Fig. 3d). We obtain additional information about this state from the PL spectra and estimate the dipole moment of this new 'dark' interlayer state to be $d_{dark} = 0.84 \pm 0.14$e · nm (SI Fig. 14).

We suggest that the new interlayer 'dark' state below $IX_1$ is associated with spin- or momentum- forbidden interlayer excitons as depicted in Fig. 3c. It was recently shown that spin-selection rules prohibiting scattering of such a state can be lifted under a strong electric field due to a Rashba-like effect[52] or under strong translational disorder introduced by the presence of molecules[46]. Alternatively, the state could be related to recently observed quadrupolar excitons[53]. The nature of this dark state deserves further study.

## Discussion
The electric field achieved via hybrid molecular gating roughly doubles the limit achievable with dielectric gates. In electric fields of up to 0.35 V nm$^{-1}$ (Fig. 2a, e) we observe, in addition to the well-known coupling between the excitons $IX_1$ and $X_B$[16,17,20]: 1) a new interlayer exciton $IX_2$ hybridizing with the $X_A$ exciton at high fields, and 2) signatures of coupling between a dark interlayer exciton and $X_A$.

To further examine the capabilities and limitations of the molecular gating technique, we applied our simple model to different combinations of TMD bilayers, heterostructures, and molecules. For TMDs, we choose combinations of the four most common materials, MoS$_2$, MoSe$_2$, WS$_2$, and WSe$_2$[50]. We expose these TMDs to four different cases of molecular doping (Fig. 4a). The first two cases are the same as considered above, F$_4$TCNQ or CN6-CP molecule on top of the TMD bilayer and SiO$_2$ below. In addition, we consider a donor n-dopant Benzyl Viologen (BV$^0$)[54] at the bottom and an acceptor CN6-CP at the top. To study the limits of our technique, we analyze the sample with CN6-CP in combination with one of the strongest organic electron donors (OED) reported – Me-OED, which has a high doping efficiency and a small surface area[55,56].

We find that the maximum achievable electric field depends both on the type of 2D material used as well as on the chosen molecule (Fig. 4b). Interestingly, switching from F$_4$TCNQ to a stronger acceptor CN6-CP does not necessarily increase the electric field strength. This is because in that case it is limited by the Fermi level entering the conduction band of the TMD. A combination of Me-OED and CN6-CP produces an electric field strength above 0.5 V nm$^{-1}$, tripling the solid-

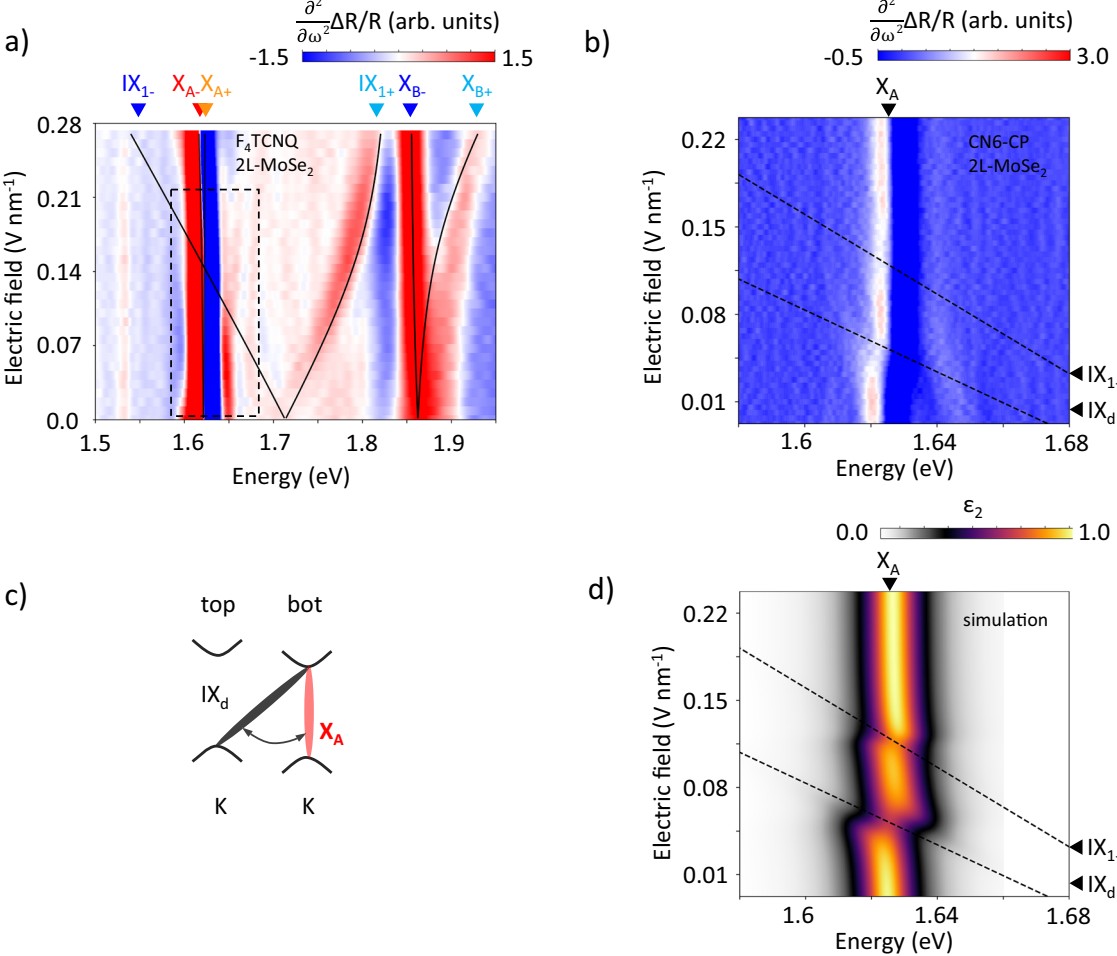

**Fig. 3 | 2L-MoSe₂ in a strong electric field. a** Map of the second derivative of reflectivity contrast for 2L-MoSe₂. Gray lines show modeled positions for $X_A$, $X_B$ and $IX_{1-}$ excitons. **b** Zoomed in region of the second derivative of reflectivity contrast around $X_A$ exciton for sample 2. Dashed lines show predicted positions for $IX_{1-}$ and dark interlayer exciton $IX_d$. **c** The proposed schematic of dark interlayer exciton. **d** Simulated absorption where $IX_{1-}$ and $IX_d$ states cross $X_A$ with coupling strengths of 5 and 10 meV.

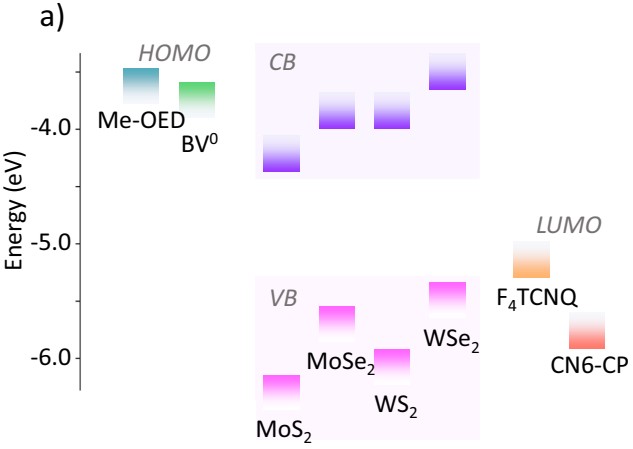

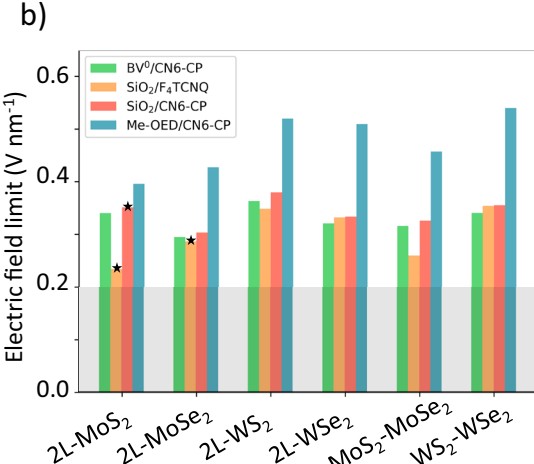

**Fig. 4 | Estimation of the maximum electric field produced by molecular double gating for several TMD/molecule combinations. a** Energy alignment of conduction/valence bands for TMDs and LUMO/HOMO levels of molecules relative to the vacuum energy. **b** Maximum electric field achievable in various TMD structures double-gated via various molecular layers. For heterostructures, the stacking order is top to bottom. Gray box corresponds to an electric field of 0.2 V nm⁻¹, achievable via dielectric gates. The star symbols indicates cases experimentally tested in this work.

state limit (details in SI note S5). Ultimately, the magnitude of the electric field is limited by the doping efficiencies of the considered organic molecules.

We anticipate several potential avenues for the application of our results. First, the ability to control the molecular density in situ at cryogenic temperatures in nanofabricated devices may prove useful for the emerging field of organic-inorganic 2D heterostructures. Second, our work indicates that TMD heterostructures functionalized with donor/acceptor species from the top and the bottom (Fig. 4) can be considered novel optical materials with the characteristic absorption peak tunable over a large part of the visible spectrum. To enable potential applications in such materials in, e.g., LED devices, it would be interesting to investigate direct chemical synthesis routes for such heterostructures[57]. Third, the response of an exciton to the out-of-plane electric field indicates its out-of-plane character and can be used, in principle, for the "fingerprinting" of excitonic species[58,59]. As we have shown, the field response of some states (e.g., $X_A$) is only resolvable in high enough electric fields. Therefore, the application of high electric fields may enable a more detailed identification of various excitonic species with debated character. Fourth, the state $IX_2$ reported here is an attractive candidate to transmit information in excitonic circuits[60,61]. This state is normally dark and should have a very long lifetime, enabling its propagation over long distances. The information encoded in it could be "written" or "read out" in the regions of the circuit exposed to a high electric field. In those areas, the state is brought into an energetic resonance with an intralayer exciton, thereby increasing its coupling to light.

## Methods

### Device fabrication
Samples were prepared using a PDMS dry stamping method and transferred onto hBN directly exfoliated onto a 285 nm $SiO_2$/Si chip[62]. Contacts were made using electron beam lithography (EBL) followed by thermal evaporation of Cr/Au (3 nm/70 nm). All samples were cleaned by AFM "nano-squeegee" (60 nN force) to clean the surface and improve the contact with molecules[63,64].

### In situ evaporation
In our technique, we place a small amount of organic acceptors $F_4TCNQ$ (Sigma-Aldrich, amount < 1 mg) or CN6-CP onto an evaporation coil fabricated on a 285 nm $SiO_2$/Si chip. The coil is made using EBL with the same parameters as the contacts on the sample. The coil resistance is 60 Ohm, and the design is similar to Ref. 65. This chip is loaded into our optical cryostat right next to the 2L-TMD. To evaporate a controlled dose of molecules, we apply a short voltage pulse to the evaporator coil (Fig. 1b, d, $V_{EVAP}$, duration is selected between 1s and 3s), heating the molecules above their melting temperature. During the heating process, the temperature of the 2L-TMD remains virtually unchanged (details in SI Fig. 10). The evaporation chamber is sealed inside the inner heatshield of the cryostat to avoid contamination. This in situ evaporation approach has multiple advantages. First, the density of molecules can be adjusted during the experiment without heating the device. Second, evaporation at cryogenic temperatures solves the problem associated with molecules agglomerating into clusters, which occurs during room-temperature deposition[33]. Finally, we avoid the exposure of a thin molecular layer to the ambient environment.

### Optical measurements
We use a home-built confocal PL/reflectivity setup at cryogenic temperature (4K). PL measurements were done using a 532 nm continuous wave laser. Reflectivity measurements were carried out using a broadband pulsed supercontinuum laser source (Super K, 400–1000 nm). All measurements employed a SLWD Nikon objective (0.4 NA, 22 mm WD) and an Andor spectrometer (300 and 600 lines/mm

gratings). To extract the positions of excitonic resonances from Fig. 2a, we used Kramers–Kronig relations to model the dielectric function (see SI note S3 for details).

## Data availability
The data generated in this study have been deposited in the Zenodo database under accession code (https://doi.org/10.5281/zenodo.16951134).

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

## Acknowledgements

S.K., K.G., A.M.K., S.P. and K.I.B. acknowledge the Deutsche Forschungsgemeinschaft (DFG) for financial support through the project GZ: BO 5142/4-1 and Collaborative Research Centers TRR 227 (project B08), SfB 951 (project B15) and SfB 1772 (project B01), as well as the Federal Ministry of Education and Research (BMBF, 05K2022 – ioARPES). J.S., Q.C. and S.E. acknowledge financial support from the DPG via SfB 1772 (project C01). D.C., M.S. and A.K. acknowledge financial support from the DPG via SfB 1772 (project A03). S.K. acknowledges useful conversations with Nele Stetzuhn, Ben Weintrub, Denis Yagodkin, Georgy Gordeev, and Theresia Knobloch.

## Author contributions

S.K., K.G. and K.I.B. conceived the project. S.K. and K.G. designed the experimental setup and developed the in situ evaporation technique. S.K., A.M.K. and S.P. prepared the samples and performed the optical measurements. J.S., Q.C. and S.E. synthesized and characterized the

CN6-CP molecules. K.W. and T.T. grew the hBN crystals. D.C., M.S., A.K., developed a theory for excitons. S.K. performed electrostatic simulations. S.K. and S.P. analyzed the data. S.K. and K.I.B. wrote the manuscript with input from all co-authors.

## Funding

## Competing interests
The authors declare no competing interests.
