## [Peer Review File · Nature Communications]

Revealing hidden interlayer excitons in 2D bilayers via hybrid molecular gating

Corresponding Author: Professor Kirill Bolotin

Version 0:

Reviewer comments:

Reviewer #1

(Remarks to the Author)

The authors have revised the manuscript in response to previous comments and present improved agreement between their model and the experimental data. While the updated figure shows a visibly better match between data points and model predictions for the XA^- and $IX2^-$ peaks, the response does not clearly explain how this improvement was achieved. Which specific aspects of the fitting procedure were modified during the revision? Were certain model parameters adjusted, or was the fitting method itself changed or refined?

Regarding Referee #2's concern: could the authors clarify how they distinguish whether a given excitonic peak is a phonon replica or a distinct excitonic state? What specific experimental signatures or criteria support this identification?

Subject to clarification of these points, I find the manuscript significantly improved and suitable for publication in Nature Communications.

Reviewer #2

(Remarks to the Author)

In the response letter, I still find it hard to understand the calibration of the electric field. The authors should make efforts to explain clearly the parameters in equation 1 in the main text, for instance, σ_t and σ_b . Are they constants, or dependent on the V_g ? Does the quantity " $\sigma_t - \sigma_b$ " change for each evaporation?

In addition, the slopes indicated by the black dashed lines in Fig. S3 (see attached file) are different. The authors should explain.

Version 1:

Reviewer comments:

Reviewer #2

(Remarks to the Author)

I have no more comments.

We thank the Reviewers for their comments. Below, we present point-by-point response to all comments and highlight changes to the manuscript. Overall, we clarified the description of excitonic peak fitting procedure and the approach to field calibrations. We hope that these changes improve the manuscript making it acceptable for publication.

Reviewer #1 (Remarks to the Author):

The authors have revised the manuscript in response to previous comments and present improved agreement between their model and the experimental data. While the updated figure shows a visibly better match between data points and model predictions for the X_{A^-} and IX_{2^-} peaks, the response does not clearly explain how this improvement was achieved. Which specific aspects of the fitting procedure were modified during the revision? Were certain model parameters adjusted, or was the fitting method itself changed or refined?

We agree, this was indeed unclear. To describe the small (10 meV) redshift of the intralayer exciton X_{A^+} (that is only very weakly coupled to IX_{2^+} and should only shift very weakly due to field-tunable interexcitonic hybridization), we accounted for small but well-established effect, the quadratic Stark effect of interlayer excitons. Under applied field F_z , the energy of intralayer excitons shifts by $\Delta E = -\beta_z F_z^2$, where β_z is exciton polarizability [Roch *et al.*, *Nano Lett.* 18, 1070–1074 (2018), Sponfeldner *et al.*, *Phys. Rev. Lett.* 129, 107401 (2022)]. Qualitatively, this effect appears because of field-induced shifting of excitonic wavefunction in the out-of-plane direction by the applied field; the effect is applied to X_{A^+} , X_{A^-} , X_{B^+} and X_{B^-} intralayer excitons.

In SI Figure S8 (also below), the black line is a fitted position of X_{A^+} without this effect being included, while the orange line accounts for the quadratic Stark effect. The data (points) are better matched by this model.

From this fitting we obtain $\beta_z = 3.59 \pm 0.42 \text{ D nm V}^{-1}$.

For completeness, we compared this value with previous reports:

Ref	Sample	β_Z ($D \cdot \text{nm} \cdot V^{-1}$)
Roch et al., Nano Lett. 18, 10701074 (2018)	1-L MoS ₂	0.78 ± 0.1
Sponfeldner et al., Phys. Rev. Lett. 129, 107401 (2022)	2-L MoS ₂	6.4
Lei et al., Nature Comm. 14, 5314 (2023)	3-L BP	4.35 ± 0.37
Lei et al., Nature Comm. 14, 5314 (2023)	4-L BP	17.46 ± 1.23
This work	2-L MoS ₂	3.59 ± 0.42

We see the obtained β_Z is smaller than what is measured by Sponfeldner et al. for the same system. We believe that this discrepancy stems from i) not taking into account the hybridization with IX₂ in that excellent work and ii) stronger overall field in our case. Our work demonstrates that the main contribution to the shift of X_{A-} is not polarizability but the hybridization with IX₂₋.

To clarify this, we referred to polarizability and its fitted value in the main text.

Regarding Referee #2's concern: could the authors clarify how they distinguish whether a given excitonic peak is a phonon replica or a distinct excitonic state? What specific experimental signatures or criteria support this identification? Subject to clarification of these points, I find the manuscript significantly improved and suitable for publication in Nature Communications.

Our key argument is that excitonic replicas do not hybridize with other excitonic states and appear shifted by a known phonon energy below the "original" excitonic states. To illustrate this, we use an example of hBN encapsulated 2L-MoS₂ with top and bottom graphene gates (Fig. below, a) where we believe that the state IX_? (red in panel b)) is a replica of IX₂. That state shows a field-splitting behavior expected for interlayer excitons and has been interpreted as IX₂ before. If that were the case, however, IX₂₋ would hybridize with X_{A-} and produce ≈ 20 meV splitting of this state at fields as small as 0.1 V/nm, and > 100 meV at the highest field value. Simulations below show that scenario (panel c, the energy of IX₂ at zero field is assumed 2.07 eV, red line shows the shift of X_{A-}). Our data do not follow any such expectations (Panel b) and the X_{A-} remains almost field independent up to large fields. Second, the energy difference between IX_? and IX₂ as reported in the main text, about 100 meV, is close to the energy difference between IX₁ and its phonon replica (Zhao et al., Phys. Rev. B 105, L041409, 2022). It also matches the theory expectations Dery et al, Phys. Rev. B 92, 125431 '2015. Therefore, the state IX_? is likely a phonon replica of the exciton IX₂ reported in the manuscript.

Reviewer #2 (Remarks to the Author):

In the response letter, I still find it hard to understand the calibration of the electric field. The authors should make efforts to explain clearly the parameters in equation 1 in the main text, for instance, σ_t and σ_b . Are they constants, or dependent on the V_g ?

We agree with the Reviewer that the electric field calibration explanation is not completely clear. In this reply, we will show three approaches to confirm it: *i*) electrostatic modelling, *ii*) excitonic shifts, and *iii*) by comparing with existing literature on the same material.

Our system is modelled as a TMD kept at zero potential between a layer of charge just above it (density σ_t , charge on the molecules), the layer of charge just below it (density σ_b , trapped in SiO₂) and the layer of charge on a Si backgate kept at potential V_G . The field-dependent measurements are carried out in the regime of zero free carrier density in the TMD – this is confirmed by the absence of the charged exciton signal (the full plotted range of the device #2 in Fig. 2a, and -80V to 45 V for device #1 in SI Fig. 6). In that regime, the field inside the TMD is given by Eq. 1 of the main text:

$$F_Z = \frac{1}{2\varepsilon_0\varepsilon_{TMD}}(\sigma_T - \sigma_B - V_G C_G) \quad (1)$$

This equation is the simplified form of the complete nonlinear equations for the charge neutrality case (SI note 1). This equation can be understood as superpositions of fields produced by the gate, top molecules and bottom charged. The model based on Eq. 1 is widely accepted in the literature. For example, it is used to calibrate the field in doubly-gated TMD transistors [e.g., *Leisgang et al., Nat. Nanotechnol. 15, 901 (2020)*]. If we replace σ_T with $V_T C_T$, and σ_B with $V^0 C_G$, we will obtain Eq. S7 of that reference (we will discuss later the meaning of voltages V_T and V^0).

Eq. 1 can be further simplified under the assumption of charge conservation:

$$\sigma_t + \sigma_b + V_G C_G = 0 \quad (2)$$

Then, using $\sigma_B = V^0 C_G$, we rewrite Eq. 1 as:

$$F_Z = \frac{\varepsilon_{SiO_2}}{\varepsilon_{TMD} d_{SiO_2}}(V_G + V^0) \quad (3)$$

This equation is used to convert V_G to field. We first identify the point of $F_Z=0$ in our gate-dependent sweeps which is signaled by zero splitting between IX₁₊ and IX₁₋. In Fig. 2a in the main text, for instance, this happens at $V_G=-86.6$ V. This allows us to determine $V^0=86.6$ V. We next assume the dielectric constant and the thickness of SiO₂ layer (3.8 and 285 nm, respectively), and use well-accepted dielectric constant for TMDs, 6.8 [*Laturia, A. et al., npj 2D Mater. Appl. 2, 6 (2018)*], and make the assumption that V^0 does not depend on V_G during a single direction sweep, based on *Ju, L. et al., Nature Nanotech. 9, 348-352 (2014)*, *Roch, J. et al., Nature Nanotech. 14, 432-436 (2019)*. Finally, Eq. 2 with all known parameters is used to convert V_G to F_Z . For example, applying this formula to extract the highest electric field in the device #2 (Fig. 2a in the main manuscript), we get: $F_Z^{max} = \frac{3.8}{6.8 \cdot 285 \text{ nm}}(90 + 86.6 \text{ V}) = 0.346 \text{ V nm}^{-1}$

In the second approach, we utilize the field-induced shifts of interlayer excitons to confirm the field calibration independently. We fit the field-dependent energy positions of IX₁₊ and IX₁₋ following *Lorchat et al., Phys. Rev. Lett. 126, 037401 (2021)* (Eq. 9, 10 from the SI, also figure below). The model uses the dipole moment of IX₁ unperturbed by interactions with intralayer excitons and the strength of excitonic interactions as fitting parameters. From that model, we obtain the dipole moment 0.576 e·nm for the sample in the main text. This value is close to the literature value of MoS₂ interlayer distance $d \approx 0.62$ nm [*Deilmann, T. et al., Nano Lett. 18, 2984 (2018)*, *Sun, J. et al., ACS Nano 16 1936 (2022)*]. This is what we expect, given the density function calculations predicting the electronic wavefunction of interlayer excitons localized on the Mo atom of one layer and the hole

wavefunction on the Mo atom of another layer. Since interexcitonic splitting is field-dependent, matching of the two numbers indicates the accuracy of our field calibration. In a simplified language, one can say we obtain the field from the energy shift of interlayer excitons (ΔE) unperturbed by interaction with intralayer excitons (IX_{1+}^0 , dashed line in the Fig. below) via $F_Z = \frac{\Delta E}{d}$. Applying this formula to IX_1 we get for the measured range $\Delta E = 200 \text{ meV}$, and:

$$F_Z^{max} = \frac{200 \text{ meV}}{0.576 \text{ e}\cdot\text{nm}} \approx 0.347 \text{ V nm}^{-1}.$$

Finally, we can directly compare the splitting of excitonic states and corresponding electric field values between our and previous works.

In *Leisgang et al., Nat. Nanotechnol. 15, 901 (2020)* the splitting between XB_+ and XB is 50 meV at their highest electric field value 0.206 V nm^{-1} . The same splitting in our work is achieved at the electric field value 0.209 V nm^{-1} . Two results match closely.

In *Peimyoo et al., Nature Nanotechnol. 16, 888–893 (2021)*, the highest electric field is smaller and the sample stays in the regime of linear IX_1 splitting. At the highest electric field of 0.071 V nm^{-1} , IX_{1+} and IX_{1-} split by 69 meV. Comparing to our work, the same splitting is achieved at 0.066 V nm^{-1} , or 0.069 V nm^{-1} when using the same dielectric constant for TMD ($\epsilon_{TMD} = 6.5$). Again, the results match closely.

The devices used by Leisgang et al. and Peimyoo et al. employ top and bottom hBN gates and should not have any issues with field calibration.

We added a condensed version of this reply to the SI.

Does the quantity " $\sigma_t - \sigma_b$ " change for each evaporation?

The charge densities σ_t and σ_b are determined from optical spectroscopies of molecular gated devices vs. gate voltage. Two signatures that we follow, are: i) the gate voltage where $F_Z = 0$, signaled by zero splitting between IX_{1+} and IX_{1-} , and ii) the gate voltage where TMD Fermi energy enters conduction band, and we start observing trions.

First, the charge density inside SiO_2 traps σ_b , corresponds to the voltage where $F_Z = 0$, as can be seen from Eq. 9:

$$F_Z \approx \frac{\epsilon_{SI02}}{\epsilon_{TMD} d_{SI02}} (V_G + V^0)$$

In Fig. 2a in the main text, for instance, this happens at $V_G = -86.6$ V, determined by fitting of IX_{1+} and IX_{1-} resonances. This allows us to determine $V^0 = 86.6$ V, and corresponding $\sigma_b = V^0 C_G = 6.4 \cdot 10^{12} \text{cm}^{-2}$. It is considered to be constant within a single voltage sweep, and within a single evaporation. On the other hand, values for σ_b between different evaporations, as experiments demonstrate, can slightly change (SI Fig. 3).

To find σ_t , gate voltage dependent charge density inside molecules, we follow the second spectroscopic signature – appearance of the trions. As long as TMD is kept neutral, the charge conservation condition holds, and carrier density inside of the top molecular layer σ_t :

$$\sigma_t = -\sigma_b - V_G C_G$$

At the same time σ_t depends on the difference between the TMD Fermi energy (E_F) and the molecules' LUMO energy. In the model that we use (Eq. 6):

$$\sigma_t = C_{mol}(E_F - E_{LUMO})$$

here E_{LUMO} is the lowest unoccupied molecular orbital energy of the molecule, that is taken from a literature ($E_{LUMO} = -5.3$ eV for $F_4\text{TCNQ}$) or the cyclic voltammetry measurement ($E_{LUMO} = -5.94$ eV for CN6-CP , SI Fig. 12). C_{mol} is the effective geometrical capacitance of the molecular layer.

σ_t reaches its maximum value once E_F enters the conduction band, and becomes nearly static due to high DOS. Further increase of V_G mostly contributes towards charging TMD. Thus, experimentally measured voltage V_G where trions just start to appear determines E_F , σ_t^{max} and C_{mol} .

For different molecular coverages, the cut-off voltage changes, modeled by the changes of the effective capacitance C_{mol} and resulting σ_t^{max} . For the device #2, at the complete molecular coverage $\sigma_t^{max} = 13.08 \cdot 10^{12} \text{cm}^{-2}$. For the device #1, one can see the evolution of the trion resonance peak with each evaporation in SI Fig. 3 and extracted σ_t^{max} in main text Fig. 1e. At the complete molecular coverage, we extract $C_{mol} = 17.9 \text{mF/m}^2$, which corresponds to a molecule/TMD distance of ~ 1 nm.

We have modified according SI note S2: “Top and bottom molecular layers charge densities determination”.

In addition, the slopes indicated by the black dashed lines in Fig. S3 (see attached file) are different. The authors should explain.

The slopes are different for that case because the condition of TMD charge neutrality used to obtain Eq. 1-3 breaks down for this sample at low molecular coverage shown here. Note that charged exciton absorption resonance below X_A (trions) appears for low molecular coverages (e.g. already for $V_G > -75$ V for sample without molecules “clean”, and e.g. for $V_G > -20$ V for partial molecular coverage “evap 2”). This means that there are free carriers inside the TMD for these voltages. These carriers screen externally applied field, making it smaller, and leading to the decreased slopes pointed out by the reviewer. This observation further highlights the need of high molecular density to obtain large fields.

We finally note that while these screening effects cannot be accounted by naïve electrostatics (Eqs. 1-3), they are captured by a detailed model accounting for quantum capacitance and TMD density of states (SI note 1). Below, we plot the positions of IX_{1+} and IX_{1-} (dashed lines) predicted by that model in the regime of finite carrier density obtained from the trion position. We obtained reasonable match to the experimental data even in that case.

For the device in this figure, there is no molecules evaporated on the surface, and trions can be seen for the majority of the voltage range, -75 to 80 V. This means that the Fermi energy of TMD is in the conduction band, and the main effect of applied voltage is change in carrier density. Interlayer exciton IX_1 in this case never comes into resonance with X_A . Whereas, for the samples with full molecular coverage, this happens at ≈ -45 V (Fig. 2a), and trions are absent in the measured voltage range.

In the response letter, I still find it hard to understand the calibration of the electric field. The authors should make efforts to explain clearly the parameters in equation 1 in the main text, for instance, σ_t and σ_b . Are they constants, or dependent on the V_g ? Does the quantity " $\sigma_t - \sigma_b$ " change for each evaporation?

In addition, the slopes indicated by the black dashed lines in Fig. S3 are different. The authors should explain.